# Urine Metabolome Dynamics Discriminate Influenza Vaccination Response

**DOI:** 10.3390/v15010242

**Published:** 2023-01-14

**Authors:** Tori C. Rodrick, Yik Siu, Michael A. Carlock, Ted M. Ross, Drew R. Jones

**Affiliations:** 1Metabolomics Core Resource Laboratory, NYU Langone Health, New York, NY 10016, USA; 2Center for Vaccines and Immunology, University of Georgia, Athens, GA 30602, USA; 3Department of Biochemistry and Molecular Pharmacology, NYU Langone Health, New York, NY 10016, USA

**Keywords:** metabolomics, influenza, vaccine, LCMS

## Abstract

Influenza represents a major and ongoing public health hazard. Current collaborative efforts are aimed toward creating a universal flu vaccine with the goals of both improving responses to vaccination and increasing the breadth of protection against multiple strains and clades from a single vaccine. As an intermediate step toward these goals, the current work is focused on evaluating the systemic host response to vaccination in both normal and high-risk populations, such as the obese and geriatric populations, which have been linked to poor responses to vaccination. We therefore employed a metabolomics approach using a time-course (*n* = 5 time points) of the response to human vaccination against influenza from the time before vaccination (pre) to 90 days following vaccination. We analyzed the urinary profiles of a cohort of subjects (*n* = 179) designed to evenly sample across age, sex, BMI, and other demographic factors, stratifying their responses to vaccination as “High”, “Low”, or “None” based on the seroconversion measured by hemagglutination inhibition assay (HAI) from plasma samples at day 28 post-vaccination. Overall, we putatively identified 15,903 distinct, named, small-molecule structures (4473 at 10% FDR) among the 895 samples analyzed, with the aim of identifying metabolite correlates of the vaccine response, as well as prognostic and diagnostic markers from the periods before and after vaccination, respectively. Notably, we found that the metabolic profiles could unbiasedly separate the high-risk High-responders from the high-risk None-responders (obese/geriatric) within 3 days post-vaccination. The purine metabolites Guanine and Hypoxanthine were negatively associated with high seroconversion (*p* = 0.0032, *p* < 0.0001, respectively), while Acetyl-Leucine and 5-Aminovaleric acid were positively associated. Further changes in Cystine, Glutamic acid, Kynurenine and other metabolites implicated early oxidative stress (3 days) after vaccination as a hallmark of the High-responders. Ongoing efforts are aimed toward validating these putative markers using a ferret model of influenza infection, as well as an independent cohort of human seasonal vaccination and human challenge studies with live virus.

## 1. Introduction

Influenza, flu, is a viral infection affecting the respiratory system with two major subtypes, influenza A virus (IAV) and influenza B virus (IBV), contributing to human disease. The virus is highly contagious and airborne, with symptoms ranging from mild to deadly [1]. Both IAV and IBV contribute to seasonal infections, while the pandemic strains typically arise from the IAV clade. In recent years, the threat of pandemic has become more acknowledged, but seasonal (epidemic) influenza is still associated with significant morbidity and mortality worldwide, with estimates of an average of 389,000 annual deaths [2] between 2002 and 2011. Therefore, IAV represents a significant public-health issue, and further work is needed to improve the prevention, surveillance, diagnosis, and treatment strategies to better understand the molecular underpinnings of the immune response to IAVfollowing infection and vaccination.

Infection by IAV or IBV begins with targeting of the epithelial cells of the respiratory system by viral hemagglutinin (HA), which mediates cell entry, trafficking to the endosomes, and, ultimately, import to the cell nucleus, where the transcription of cRNA and vRNA takes place. The expression of the viral protein and RNA activates the innate and adaptive immune responses, leading to overt symptoms of infection and changes in cellular metabolism [3,4,5]. IAV strains consist of different combinations of the two surface proteins found on the virus, hemagglutinin (HA) and neuraminidase (NA). There are 18 subtypes of hemagglutinin (H1-H18) and 11 subtypes of neuraminidase (N1–11) [1]. NAs make up approximately 10–20% of influenza surface proteins, while HAs make up approximately 80–90% of surface proteins, partly explaining why most vaccine designs target the HA protein. IBV is similar, and strains are similarly grouped by HA, but divided into two different lineages (B/Victoria or B/Yamagata) instead of subtypes. Currently, H1N1, H3N2, B/Victoria, and B/Yamagata are all co-circulating seasonally in humans. One driver of the need for an annual vaccination is antigenic drift, in which mutations [6,7,8,9] in the genes that code for the antibody binding site, reducing binding recognition by the existing host antibodies. Antigenic shift during co-infection further generates the potential for novel immune-evading combinations of viral glycoproteins. With this reassortment during co-infection, IAV has the potential to generate hundreds (256) of unique genetic combinations of the two parental strains.

Because of these challenges, influenza vaccination can lack specificity and efficacy [10]. Currently, the seasonal vaccine strains are selected based on statistic modelling [11] based on the observed configurations and other pre-season metrics. Most seasonal vaccines are trivalent or quadrivalent [12], and the vaccination efficacy can vary dramatically from year to year, while the responses can be population-dependent. This changing landscape of strain selection and seasonal vaccination highlights the high-risk populations who exhibit poor responses to vaccination. The 2009 H1N1 (IAV) pandemic revealed the severity of obesity (BMI > 30) as a leading risk factor for more severe infections and higher mortality [13]. The geriatric population (>65 years old) has also long been recognized [14] as having less robust responses to influenza vaccination, leading to the use of high-dose or adjuvanted vaccine designs for these populations. For each of these high-risk populations, the mechanism of reduced vaccine efficacy is multi-factorial and incompletely understood.

While research suggests that there is a link between high-risk populations and seroconversion following influenza vaccination, not much is known about the underlying metabolic mechanisms. Further characterization at the molecular level is also needed to understand what constitutes a robust immune response to vaccination and link these changes to the vaccine design and function of the host immune system. The two high-risk populations of interest in the current study both share systemic changes in their overall metabolism. Therefore, we hypothesized that the metabolic profiles of subjects undergoing influenza vaccination may reveal seroconversion-dependent changes in their systemic metabolism, which could aid in characterizing, identifying, and predicting the biochemical processes mediating a robust immune response to vaccination. As the end products of the cell regulatory process, metabolism is generally considered to be the most sensitive of the omics disciplines at detecting differences associated with the phenotype [15,16,17] and is playing an increasingly impactful role in the investigation of novel mechanisms of pathophysiology. To begin addressing these questions, we examined a cohort of healthy adults undergoing annual influenza vaccination to identify potential metabolite markers that may be linked to effective response to vaccination among high-risk groups through metabolomic analysis. Our cohort comprised cross-sectional sampling with respect to age, sex, BMI, and other demographic factors so that we could associate metabolic changes with the vaccine response. We aimed to use seroconversion to the vaccine strains as a proxy of protective immune response to influenza vaccination with the goal of then correlating the seroconversion score to various time-dependent metabolic changes in the general cohort, as well as in the obese and geriatric subsets.

## 2. Methods

### 2.1. Vaccine Cohort

The current metabolomics study utilizes a 2019–2020 cohort of urine samples from the University of Georgia (*UGA4*), which were acquired from subjects receiving split, inactivated Fluzone^TM^, as previously described [18,19]. The study procedures, informed consent, and data collection documents were reviewed and approved by the Western Institutional Review Board and the Institutional Review Boards of the University of Pittsburgh and the University of Georgia. All subjects were recruited from the Athens, Georgia geographic region, including the University of Georgia. Background demographic data on the population were acquired from the Centers for Disease Control (CDC), Athens-Clarke County Unified Government, and United States Census Bureau. Subjects were excluded from the batch assignment and sample processing if one or more of the 5 urine sample time points were missing or unavailable. No other exclusion criteria were applied. The final study consisted of 179 unique subjects, each with five time points for a total *n* = 895 urine samples analyzed. HAI assays were carried out and seroconversion score were obtained as previously described [18,19].

### 2.2. Batch Design and Quality Control

To minimize the batch effects, the subjects were randomized into 31 technical batches, and all the sample time points for each subject were analyzed together. Each batch, therefore, contained 6 subjects, and each of their 5 time points for 30 total samples per batch, excluding the final batch. Urine samples were extracted as described below, and the order of acquisition was randomized to minimize the sequence (within-batch) effects. Each LC-MS sequence contained several control blocks of a standard cocktail (blank extraction buffer, with no sample but containing internal standards) that were extracted alongside each batch. The control block consisted of a blank control followed by a standard and another blank. The control blocks were injected at the start of the run, in-between every 6 samples, and at the end of the run, so that the instrument performance could be monitored throughout. These control injections were used to assess the data quality for each batch and measure the instrument variance, carry over, and column stability. Analytical blank injections were further used to define the blank threshold for peak detection in each batch. A large volume (2 L) of extraction buffer and 80% methanol for the internal blank controls was generated before the study to be used in the technical aliquots for all the subsequent technical batch processing.

### 2.3. Extraction of Metabolites from Urine

For each batch, the appropriate samples were removed from −80 °C storage and transferred to wet ice and thawed. An aliquot of extraction buffer of 100% LCMS-grade methanol (Fisher Scientific, Waltham, MA) containing 625 nM metabolomics amino acid mix standard (Cambridge Isotope Laboratories, Inc, Tewksbury, MA.) was taken from 4 °C storage and equilibrated on dry ice for >15 min prior to sample processing. Urine samples were extracted by combining 200 µL of the sample with 800 µL of extraction buffer in 2.0 mL screw cap vials containing ~100 µL disruption beads. The tubes were homogenized for 10 cycles in a Benchmark Scientific Bead Blaster^TM^. Each cycle consisted of 20 s of homogenization at 6 m/s, followed by a 30 s pause. The homogenized samples were centrifuged at 21,000× *g* for 3 min at 4°C, and then a fixed volume of the supernatant (450 µL) was dried down using speed vacuum concentration (Thermo Fisher, Waltham, MA). Once dry, the samples were stored at −80 °C until processing. On the day of the LCMS data acquisition, the samples were reconstituted in 50 µL of LCMS-grade water, sonicated for 2 min, and centrifuged at 21,000× *g* for 3 min at 4 °C to exclude any insoluble particulates. The extracted samples were then transferred to 2 mL glass vials (Agilent, Santa Clara, CA, USA) with glass LC inserts for analysis. All the samples were stored at −80 °C after data acquisition and QC evaluation. This process was repeated for each of the 31 randomized batches.

### 2.4. LC-MS/MS with the Polar Global Metabolomics Method

The samples were subjected to an LCMS analysis to detect and quantify the putatively identified metabolites. The LC column was a Millipore^TM^ ZIC-pHILIC (2.1 × 150 mm, 5 μm) coupled with a Dionex Ultimate 3000^TM^ system, and the column oven temperature was set to 25 °C for the gradient elution. A flow rate of 100 μL/min was used with the following buffers: (A) 10 mM ammonium carbonate in water, pH 9.0, and (B) neat acetonitrile. The gradient profile was as follows: 80–20% B (0–30 min), 20–80% B (30–31 min), and 80–80% B (31–42 min). The injection volume was set to 2 μL for all the analyses (42 min total run time per injection). Each Millipore^TM^ ZIC-pHILIC column was tracked and used only for the urine samples associated with the current study.

MS analyses were carried out by coupling the LC system with a Thermo Q Exactive HF^TM^ mass spectrometer operating in the heated electrospray ionization mode (HESI). The method duration was 30 min, using a polarity switching, data-dependent top 5 method for both the positive and negative modes. The spray voltage for both the positive and negative modes was 3.5 kV, and the capillary temperature was set to 320 °C, with a sheath gas rate of 35, aux gas of 10, and max spray current of 100 μA. The full MS scan for both polarities utilized a 120,000 resolution with an AGC target of 3e6 and a maximum IT of 100 ms, and the scan range was from 67 to 1000 *m*/*z*. The tandem MS spectra for both the positive and negative mode used a resolution of 15,000, AGC target of 1e5, maximum IT of 50 ms, isolation window of 0.4 *m*/*z*, isolation offset of 0.1 *m*/*z*, fixed first mass of 50 *m*/*z*, and 3-way multiplexed normalized collision energies (nCE) of 10, 35, and 80. The minimum AGC target was 1e4, with an intensity threshold of 2e5. All data were acquired in the profile mode.

The quality control for each batch was assessed using the cocktail of isotopic amino acid standards present in both the samples and control block injections. The sample and standard chromatograms from each batch were visually compared against historic references for the expected retention time, resolution of the isomers, e.g., Isoleucine/Leucine, and the expected signal intensity for each standard. If the retention time deviated from the expectation by >0.5 min, if the column resolution was poor, or if the signal intensity was low, the instrument was cleaned and serviced as required to meet the performance benchmarks, and the samples were reanalyzed. After the data acquisition, the relative intensity of each isotopic internal standard was measured using a standard template, and the median coefficient of variation for the sequence had to be less than or equal to 15% for all the standard injections of the batch. Technical batches not meeting these criteria were reacquired after troubleshooting and instrument servicing as required.

### 2.5. Metabolomics Data Analysis

*Relative quantification of metabolites.* For each batch, the resulting Thermo^TM^ RAW files were converted to SQLite format using an in-house python script to enable downstream peak detection and quantification. The available MS/MS spectra were first searched against the NIST17 MS/MS [20], METLIN [21] and respective Decoy spectral library databases using an in-house data analysis python script adapted from our previously described approach for metabolite identification false discovery rate control (FDR) [22,23]. Here, this FDR value is reported after each metabolite’s name throughout and in Appendix A. Then, the putatively identified metabolites from all the batches with their corresponding metabolite names, accurate masses, and retention time ranges were merged together, and any duplicated metabolites names were filtered out to generate a list of metabolites with unique names. Next, the decoy hits in the resulting list were dropped, and two different FDR cutoffs (10% and 100%) were applied to the final refined metabolite list. Finally, for each sample, the peak heights for each putative metabolite hit were extracted from the sqlite3 files based on the metabolite retention time ranges and accurate masses in the above-mentioned merged metabolite list. Metabolite peaks were extracted based on the theoretical *m*/*z* of the expected ion type, e.g., [M+H]^+^, with a 15 part-per-million (ppm) tolerance and a ± 0.2 min peak apex retention time tolerance within an initial retention time search window of ±0.5 min across the study samples for each batch. The resulting data matrix of metabolite intensities for all the samples and blank controls was processed using an in-house python script, and the final peak detection was calculated based on a signal-to-noise ratio (S/N) of 3× compared to the blank controls, with a floor of 10,000 (arbitrary units). For the samples where the peak intensity was lower than the blank threshold, the metabolites were annotated as not detected and were imputed with either the blank threshold intensity for statistical comparisons so as to enable an estimate of the fold change, as applicable, or zeros for the median metabolite intensity calculation of a sample. The resulting blank corrected data matrixes obtained from the individual batches were then merged together to generate the final data matrix for all the downstream analyses. To account for the inter-batch variations and inter-patient variations, the median metabolite intensity detected for each sample was used to normalize all the detected metabolite intensities in that sample. Then, the median normalized data from all five time points for each patient were normalized to each subject’s Day 0 signal to account for inter-personal variations in the baseline. Finally, a Log2 transformation was applied to the final data matrix in order to facilitate comparisons of the up- and down-regulated metabolites. For all the group-wise comparisons, t-tests were performed using the Python SciPy (1.5.4) [24] library to test for differences and generate statistics for the downstream analyses. For the pairwise t-tests, any metabolite with a *p*-value < 0.01 was considered significantly regulated (up- or down-) for prioritization in the subsequent analyses. Heatmaps were generated by hierarchical clustering, performed based on the imputed matrix values utilizing the R library pheatmap (1.0.12). GraphPad Prism 9 (9.4.1, GraphPad Software, San Diego, CA) was used for all the volcano, line, scatter plot generation, and one-way and two-way ANOVA statistics were conducted as annotated. The univariate ROC curves were analyzed through the metaboanlyst.ca portal using classical univariate ROC curve analyses. Multivariate ROC analyses were performed by ROC-curve-based model evaluation (Tester) with the manually selected features and random forest algorithm.

## 3. Results

We employed a cohort of human volunteers (*n* = 179) recruited from the Athens, Georgia area to investigate metabolic markers of the influenza vaccine response using a time-course design. Urine samples were collected at baseline before vaccine administration (Day 0) and on four other days post-vaccination: Day 3, Day 7, Day 28, and Day 90 (Figure 1A, Appendix A). The cohort was designed so as to obtain an approximately even sample of subjects with respect to BMI, the vaccine response, age, and sex (Figure 1 B–E). The subjects were assigned to three categories based on their BMI, including normal (BMI < 25), overweight (BMI 25–30), and obese (BMI > 30). Although the number of subjects in each category was similar, the BMI distribution was not flat, with a median BMI of 28 (Figure 1F). The cohort of subjects ranged from 18 years old to 80 years old (Figure 1G), with an overall 41% female makeup. Finally, most of the cohort (86%) self-reported as Caucasian, but all the strata of BMI were represented in each demographic sampled (Figure 1H). We compared our cohort of volunteers to the background population to determine whether sampling bias could impact the study results. The catchment area had a population of 127,315 (US Census Bureau, 2021) with a median age of 28.0 years old and was 58.0% White (Athens-Clarke County Unified Government). The obesity rate of the population was estimated to be 33.9% (CDC, 2021). Our cohort was enriched in Caucasian individuals, but the BMI distribution was similar to the background. Therefore, further work is needed to determine the broader applicability of these results beyond this population. All the urine specimens were stored frozen until the metabolite extraction and LC-MS/MS-based metabolomics analysis.

Overall, we detected 15,903 putatively identified metabolites (4473 at 10% FDR) among the 895 urine study samples utilizing an MS2 spectral library search approach against the NIST17 and METLIN MS/MS spectral libraries. We carried out a relative quantification of these putatively identified metabolites across all the samples (Appendix A) and performed a semi-supervised hierarchical clustering analysis (each subject in sample order of D0–D90) to examine the overall study profiles (Figure 2A). We found that the profiles were largely homogenous, indicating no major confounding roles of random batch effects, the study design, or other systematic errors. Next, we examined the overall changes in the metabolic profiles at the group-wise level between Day 3 post-vaccination and Day 0 pre-vaccination. We used hemagglutination inhibition assay (HAI) fold change data from the pre- and post-vaccination timepoints to determine the seroconversion status and then assigned the subjects to one of three categories based on their seroconversion score: the “None” responders (<4), “Low” responders (4–7), and “High” responders (8+). Compared to each subject’s pre-vaccination baseline, in a pairwise analysis, the High-responders on Day 3 showed fewer changes in their metabolic profiles (*n* = 33 metabolites, *p* < 0.01) than the None-responders, with the Low-responders showing the most disruption (Figure 2B–D). These results support the hypothesis that robust immune responses are associated with specific changes in metabolism. Therefore, we examined the shared metabolite changes between these groups with respect to the response to vaccination on Day 3 (Figure 2E). Interestingly, when using *p* < 0.01 as the cutoff criteria, we found that there were no overlapping metabolites among all three responder groups with respect to D3. These results suggest that there are no broad metabolite markers of vaccine exposure unrelated to the vaccine response. Rather, this may indicate that the metabolic profile reflects the degree of response as None, Low, or High. To explore the possibility of overlapping the metabolites further, we lowered the criteria to *p* < 0.05 and found 38 metabolites, including Arginine, Kynurenine, Acetyl-Alanine, and D-Psicose, which may serve as markers of vaccine response aspects that are not solely related to seroconversion.

Next, we considered the metabolic changes unique to the High-responder group, as high-priority candidates (*n* = 33) for biological investigation among the overall cohort. We screened each candidate based on the significance, fold change, and manual inspection and sought to determine whether the Low-responders showed an intermediate level of change in each metabolite compared to the None- and High-responder groups. Guanine (5% FDR) had the most significant (*p* = 0.0023, one-way ANOVA) difference and was approximately 1.8-fold lower in the High-responders than the None-responders (Figure 3A). A closely related purine metabolite, Hypoxanthine (<1% FDR, *p* = 0.0440), showed a similar trend to Guanine, potentially indicating the important role of these nucleotide bases in the response to vaccination. We found that most of the metabolites unique to the High-responder group, such as Guanine and Hypoxanthine, were negatively correlated with lower metabolite levels in the High-responders compared to the None-responders. We also observed that the Low-responder group tended to match the metabolite level of the None-responders, indicating that, metabolically, the Low-responders were more similar to the None-responder group with respect to the levels of metabolites associated with the vaccine response. One exception to this trend was Acetyl-Leucine (<1% FDR), which was positively correlated with seroconversion, and the Low-responders showed an intermediate level of Acetyl-Leucine (Figure 3A). Therefore, we examined the overall correlation of these candidate markers with each subject’s seroconversion score to determine whether these markers could be used to measure the immune response to vaccination (Figure 3B). Both Guanine and Hypoxanthine showed statistically significant (*p* = 0.0091 and 0.0352, respectively) negative correlations with the subjects’ seroconversion score, while Acetyl-Leucine and the putatively identified Deuteroporphyrin IX (49% FDR) showed positive correlations (*p* = 0.0688 and *p* = 0.0024, respectively). While it only approached significance in its correlation with the seroconversion score, Acetyl-Leucine showed one of the steepest slopes at 0.033, indicating that, overall, the D3 urinary metabolite levels are only weakly predictive of the ultimate seroconversion (measured at D28). Using this candidate population of metabolite markers of the High-responders on Day 3 post-vaccination, we carried out a metabolite pathway enrichment analysis to test the hypothesis that these metabolites are related through biosynthesis (Figure 3C). We found that the Purine metabolism appeared to be significantly enriched (*p* = 0.027, raw), but this result did not survive Holm adjustment, suggesting that these metabolites may reflect the response. The set of markers unique to the None-responder group showed enrichment in Pentose and glucuronate interconversion, the TCA cycle, and other pathways, but these also did not survive Holm adjustment (Figure 3D).

Next, we examined the impacts of obesity (BMI > 30) on the differential metabolic responses to influenza vaccination. Our cohort of obese subjects showed similar proportions of High-responders (44%) and None-responders (30%). We sought to identify candidate markers of this differential response in the D3 pairwise comparison for each subject relative to their pre-vaccine baseline. We used Student’s t-test (two-tailed, equal variance, uncorrected) to prioritize the candidate differential markers and found *n* = 523 metabolites which efficiently separated the obese High-responder subjects from the obese None-responder subjects in an unsupervised hierarchical clustering analysis, according to their urinary metabolic profiles (Figure 4A). We then examined whether these markers had an overall correlation with the subject BMI. We found that the putatively annotated metabolite 3′-Hydroxyflavanone (19% FDR) was the most correlated with the subject BMI (Figure 4B) and showed an inverse relationship. Other notable metabolites included Thioguanine (2% FDR), Homogentisic acid (1% FDR), and L-Carnitine (<1% FDR). Next, we investigated whether any of these markers of differential response among the obese subjects were predictive of the D3 metabolic response. Therefore, we compared the overlapping metabolites that were significantly different between the D3 obese High-responders and the obese None-responders, as well as the D0 obese High-responders and the obese None-responders (Figure 4C). Interestingly, only one metabolite was found to overlap, which was putatively identified as D-Psicose (8% FDR), a dietary aldohexose sugar. Therefore, we examined this metabolite for its potential as a predictive marker of vaccine response among the obese population using a receiver operator characteristic curve (Figure 4D). Using baseline pre-vaccination data (D0), the urine levels of the Psicose metabolite weakly predicted (AUC of 0.68) seroconversion among the obese subjects. We then examined the metabolites that were uniquely differential between obese High-responders and obese None-responders at baseline (D0, *n* = 15) using a pathway enrichment analysis. We found that changes in Glutamate (<1% FDR) and Pyroglutamic acid (<1% FDR) led to a significant enrichment (Holm *p* = 0.0156) in the Kegg pathway “D-Glutamine and D-Glutamate Metabolism” (Figure 4E). Interestingly, when examined through an ROC analysis, Glutamic acid and Pyroglutamic acid also showed weak biomarker potential, with each having an AUC of 0.70 (data not shown). We sought to determine whether a different underlying BMI distribution between the two groups could be a confounding factor in these results, but the evaluation of the histogram for each group showed overlapping and non-significant distributions for the BMI group mean (t-test, *p* = 0.6192) and variance (F test, 0.3220) (Figure 4F). We then examined whether a differential distribution of the subjects’ ages could be a confounding factor in the analysis of the obese High-responder and obese None-responder groups. Our cohort, indeed, showed a significant difference (*p* = 0.0110) in the number of aged subjects (the mean age was increased by 8 years) among the obese None-responders (Figure 4G), potentially indicating that age is a greater relative risk factor for poor vaccine responses than obesity alone.

Based on the disproportionate role of age in predicting the obese responder status, we then examined the impact of age on the response to vaccination in the overall cohort. Unlike BMI, which was roughly evenly distributed with respect to the responder status, we found a significant age bias among the None-responders regardless of the BMI status (Figure 5A–D), but there was no correlation between the subject age and BMI in our overall cohort (*p* = 0.2988). We again used an uncorrected *p*-value (<0.01) on D3 to prioritize the candidate differential markers of the geriatric (≥65 yr age) High-responders (*n* = 9) and None-responders (*n* = 25) and found *n* = 624 differential metabolites (Figure 5E). The subjects’ profiles showed a stark clustering in this unsupervised analysis, with the largest magnitude fold changes among the up-regulated metabolites of the geriatric High-responder clade. Therefore, we again examined these candidate markers by comparing the shared metabolites between the High-responders and None-responders with respect to their Day 3 response to the vaccine vs. baseline (Figure 5F). A pathway enrichment analysis of the metabolites unique to the geriatric None-responders again showed “D-Glutamine and D-Glutamate Metabolism” due to Glutamate and Pyroglutamic acid changes, but this was not significant after Holm correction (*p* = 0.0025, Holm = 0.2108, Figure 5G). The pathway “Tryptophan Metabolism” was also enriched with four metabolites, namely 5-Hydroxyindoleacetate (15% FDR), L-Kynurenine (4% FDR), Formyl-N-acetyl-5-methoxykynurenamine (32% FDR), and 6-Hydroxymelatonin (50% FDR), but this result did not survive Holm adjustment (*p* = 0.0018, Holm = 0.1531). We then examined several metabolites associated with the None-responders in the geriatric group for correlations with seroconversion across the entire cohort (Figure 5H–K) with respect to the subject-specific changes between the pre-vaccination (D0) and D3 samples. We found statistically significant correlations (non-zero slope) between these markers, with most showing a negative correlation. One exception was 5-Aminovaleric acid, which showed a positive correlation with the seroconversion score and also showed one of the steeper slopes of the markers, which were manually interrogated. Each point increase in the seroconversion score was associated with a 3% increase in the associated level of 5-Aminovaleric acid.

Finally, we assessed the metabolites of interest from all the earlier analyses for their potential as dynamic markers of the vaccine response over the 90-day time-course of observation post-vaccination (Figure 6A–I). As a negative control, we examined the obesity-correlated metabolite 3′-Hydroxyflavanone (Figure 4B, Figure 6I). This metabolite was not previously identified as statistically significant with respect to the responder status but was significantly correlated with the subject BMI. An assessment of its urinary levels across the entire cohort of High-responders and None-responders throughout the time-course showed nearly identical traces between the two groups following vaccination, and this was indicative of the vast majority of metabolites in the overall dataset. In contrast, the purines Guanine and Hypoxanthine showed highly significant differences with respect to the High- and None-responders over time (Two-way ANOVA—row factor: days post-vaccination, column factor: responder status). Hypoxanthine was lower in the High-responders at all the time-points post-vaccination than the None-responders (*p* < 0.0001), with the metabolite levels staying fairly flat among the None-responder group. The None-responders also showed flat levels of Guanine, whereas the High-responders appeared to show oscillating levels with a return to baseline by Day 90. Several other metabolites showed similar patterns, according to which the None-responders showed flat profiles, while the High-responders showed concave down-regulation, including Orotic acid, m-Cresol, and L-Cystine (Figure 6F–H). Lactic acid (Figure 6E) showed a unique profile in that the High- and None-responders changed in opposite directions on D3 post-vaccination but were otherwise in lockstep, though this observation did not reach significance in the two-way ANOVA. The High-responders also showed higher levels of Acetyl-Leucine in the first week post-vaccination, while the None-responders showed a muted and delayed response of the same profile (Figure 6B). Notably, none of these manually assessed metabolites of interest showed a statistically significant interaction between the ANOVA factors (days post-vaccination and the responder status), possibly indicating that the overall metabolic mechanism was the same in both groups, with varying magnitudes of change. We further assessed these markers for their specificity to either the geriatric group or non-geriatric (adult) group. Some metabolites showed stronger trends in either group, but interestingly, both Guanine and Hypoxanthine were statistically significant in both groups independently, potentially highlighting their role as broad markers of the response to influenza vaccination (Appendix A).

## 4. Discussion

Our results show that the metabolome of human subjects is highly descriptive of the moment-by-moment phenotype and displays high levels of inter-individual variation that are relatively stable over several months. We accounted for this variation by comparing each subject’s post-vaccination profile with their respective baseline pre-vaccination profile in a paired fashion over time to interrogate the human metabolic response to influenza vaccination. Using this approach, our data indicate that advanced age is the highest risk factor for poor seroconversion, as assessed in our cohort, followed by high BMI. Among the geriatric subjects with a poor response to vaccination (None-responders), we found that the Tryptophan Metabolism pathway showed a trend of enrichment in several closely related Kynurenine metabolites. Interestingly, Kynurenine and its related metabolites are now well-established as regulators of the immune system, functioning primarily as immunosuppressive modulators [25,26,27,28]. Kynurenine is synthesized by the enzyme Indoleamine 2,3-dioxygenase (IDO1/2) and Tryptophan 2,3-dioxygenase (TDO). Experiments on IDO knockout mice have led to highly pro-inflammatory signals, with IDO now known to be under the transcriptional control of interferon-gamma (IFN-γ) and other cytokines [29,30].

Interestingly, two metabolites (Glutamic acid and Pyroglutamic acid) were identified as significantly altered in both of the high-risk groups evaluated in this study, including the obese subjects and the geriatric population. Additionally, we observed an aldose-sugar, putatively identified as D-Psicose (Allose), as a marker of the None-responders among the obese subjects. This was the only identified metabolite which was significantly different between the High-responders and None-responders on both Day 3 post-vaccination and at baseline pre-vaccination among the obese subjects (Figure 4C). Psicose has mixed reports in the literature regarding its immunosuppressive activity [31] but is generally regarded as safe (GRAS) and is a common low-calorie alternative sweetener found in foods. Our data on Psicose, Glutamic acid, and Pyroglutamic acid suggest the potential roles of these metabolites as predictive markers of the vaccine response. Therefore, we believe that this finding represents dietary sources of this hexose sugar, though not exclusively from additives, as the compound is naturally occurring in foods. The detection of a predictive metabolite is somewhat unexpected, because metabolism is typically thought to be highly descriptive of the state of the organism (phenotype) but not necessarily of the biological potential, though there are many counter examples [32,33,34,35,36] in biomedical science. We attempted to combine our three predictive metabolite markers in a single test and achieved a better predictive performance with an overall AUC of 0.791, suggesting a possible role of non-invasive metabolite testing in the prediction of high-risk groups for vaccine response. However, the underlying mechanism determining why these markers predict the response in obese subjects requires further investigation. Pyroglutamic acid has recently been proposed [37] as a prognostic marker for infection, and it is a key intermediate in the recycling of Glutathione, where it is thought to accumulate due to ROS stress. Interestingly, we found that Cystine, the oxidized dimer of the amino acid Cysteine and precursor of glutathione, was one of the most significantly different metabolites between the High-responders across the time-course (Figure 6H). The extracellular levels of Cystine and Glutamic acid are regulated in part by SLC7A11, which is an antiporter for these two metabolites and has been linked to Treg proliferation [38]. Similarly, Glutamic acid is recognized as an important modulator of immunity, especially in the gut [39], but the overall mechanism of this link is unclear, since it is critical for so many aspects of metabolism among lymphoid and non-lymphoid tissues.

Beyond the high-risk populations, we observed several metabolites which were more broadly correlated with high seroconversion to the vaccine. Two of these metabolites, Acetyl-Leucine and Deuteroporphyrin IX, fit our hypothesized model for a diagnostic marker of the vaccine response, showing progressively higher levels of the metabolite with higher seroconversion (Figure 3A). Although Acetyl-Leucine’s overall correlation with seroconversion was only approaching statistical significance, the Low-responder group showed a mean increase of ~15%, while the High-responder group showed a mean increase of ~30% on Day 3 post-vaccination (both significant). Deuteroporphyrin IX showed a statistically significant correlation with the seroconversion score, and while the heme degradation pathway is complex, Deuteroporphyrin IX has been linked to oxidative stress and Glutathione metabolism [40], which could be explained by an inflammatory response to the vaccine. Guanine showed a much larger mean change, with an 86% percent decrease, which was unique to the High-responder group on D3 after vaccination, while Hypoxanthine showed a 41% mean decrease. Most eukaryotic proteins are acetylated, and free Acetyl-Leucine in humans is thought to primarily derive from these post-translational modifications [41,42], potentially indicating a higher rate of protein degradation among the High-responders, but direct synthesis by Leucine N-AcetylTransferase [43] or microbial production [44] is also possible. Studies have shown that N-Acetyl-Leucine is more rapidly taken up by cells, acting as a natural pro-drug to regenerate Leucine intracellularly [45] and potentially impacting mTOR signaling. Furthermore, the acetylation of Leucine switches its import from the L-type amino acid transporter (LAT1) to the organic anion transporters (OAT1 and OAT3). A related synthetic analog, N-Acetyl-Leucine-Amide (NALA), is used experimentally to inhibit T-cell activation [46,47] through competition with Leucine import by LAT1 (SLC7A5), potentially implicating Acetyl-Leucine as a signaling molecule that functions to activate mTOR while bypassing LAT1 regulation. More broadly, we also showed that the purine metabolites Guanine and Hypoxanthine were significantly lower after vaccination in the High-responders. Hypoxanthine is a key intermediate in the purine salvage pathway and is generated by Guanine and the highly labile metabolites ATP, AMP, and Adenosine. While they are not normally detected in urine, excepting cases of high inflammation [48], the phospho-purines and purine-nucleosides have potent and complex pro- and anti-inflammatory roles [49,50]. As the downstream product of the purine salvage pathway, excreted urinary Hypoxanthine may then serve as a proxy of this complex inflammatory signaling pathway [51].

The goal of the current work was to identify candidate signals associated with human immune responses to influenza vaccination and to propose possible mechanisms for these associations for validation in future studies. A major limitation of the investigation of the human immune response was our reliance on HAI assays to measure seroconversion to the vaccine in order to define the None-, Low-, and High-responder individuals. While HAI is a long-standing metric of the immune response to generate anti-HA antibodies to block influenza receptor binding [52], HAI is only a proxy of protection, and HAI-based measures of seroconversion can be confounded by pre-immunization or cross-reactivity to closely related strains, reducing the ratio of titers between the pre- and post-vaccination timepoints. The urine samples used in this study were from a cohort of subjects who received their annual influenza vaccination in the 2019–2020 season. While the cohort was carefully constructed so as to sample across various demographic factors, as discussed, it was inherently an observational study, with the primary control condition being the baseline pre-vaccination samples. Another limitation is that our study did not account for prior immunity, immunologic imprinting (original antigenic sin) [53,54], or other factors which may complicate the apparent seroconversion of subjects. However, these issues are an active area of study [52,55] for this same cohort of subjects. Future work would benefit from human challenge studies, where the immune response can be defined as protection against symptomatic infection following live virus exposure. Another caveat is that while the urine metabolite profile is comprehensive, it is not clear which tissues generate the vaccine-response-associated metabolite signals. Urine is a convenient and non-invasive source of human samples, but further investigation is needed to define whether these markers arise from the lymphoid tissues, whole-body metabolism, or other compartments. Finally, these data demonstrate that metabolism is a powerful tool for discriminating interpersonal variations in vaccine response. Our results show that metabolism, after just three days following vaccine administration, is impacted in a such way that the ultimate immune response assessed almost one month later can be differentiated. Our ROC analyses suggests that baseline metabolic differences may even be able to predict the response to vaccination, highlighting the poorly understood role of metabolism in immunity and the need for continued work.

## Figures and Tables

**Figure 1 viruses-15-00242-f001:**
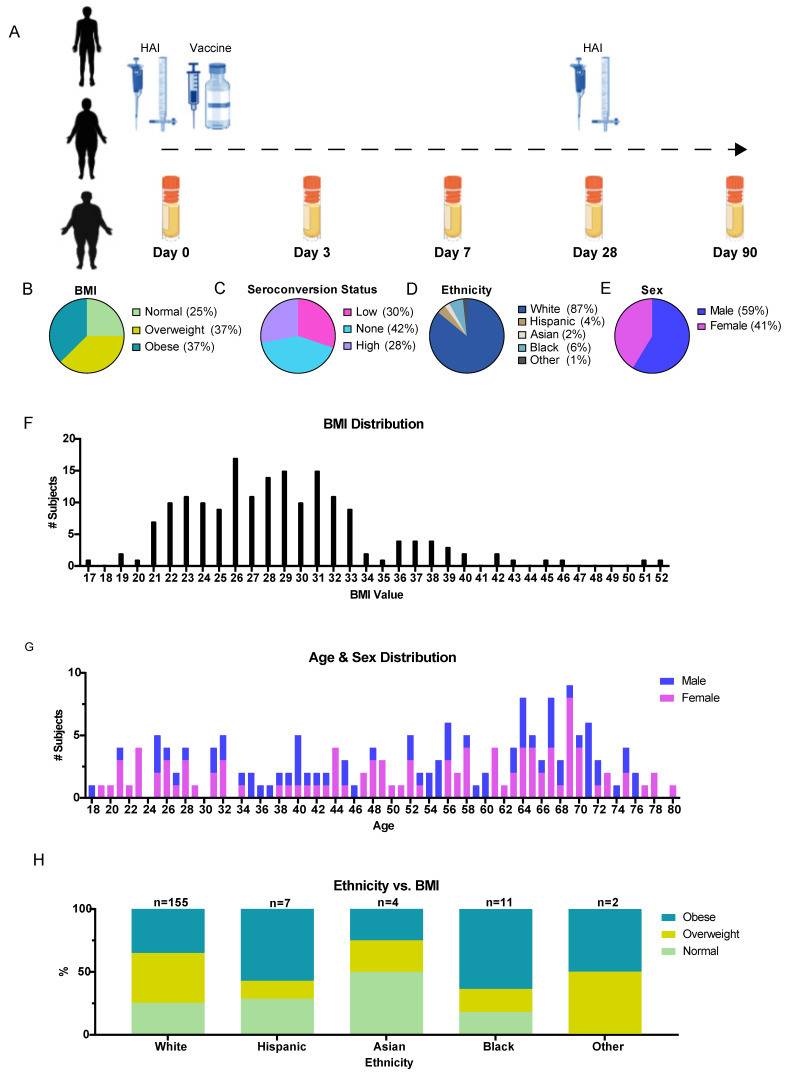
Overall design and metadata of the cohort. (**A**) Schematic representation of the time-course. Urine specimens were collected on each of 5 days for each subject. HAI was calculated using the Day 0 and Day 28 samples to determine the seroconversion status. (**B**–**E**) Proportional composition of the cohort for BMI, responder status (seroconversion), ethnicity, and sex. (**F**) Histogram of subject BMI across the vaccine cohort. (**G**) Histogram of subject age and sex across the vaccine cohort. (**H**) Proportional composition of BMI with respect to subject self-reported ethnicity. All subjects were recruited from the Athens, Georgia region.

**Figure 2 viruses-15-00242-f002:**
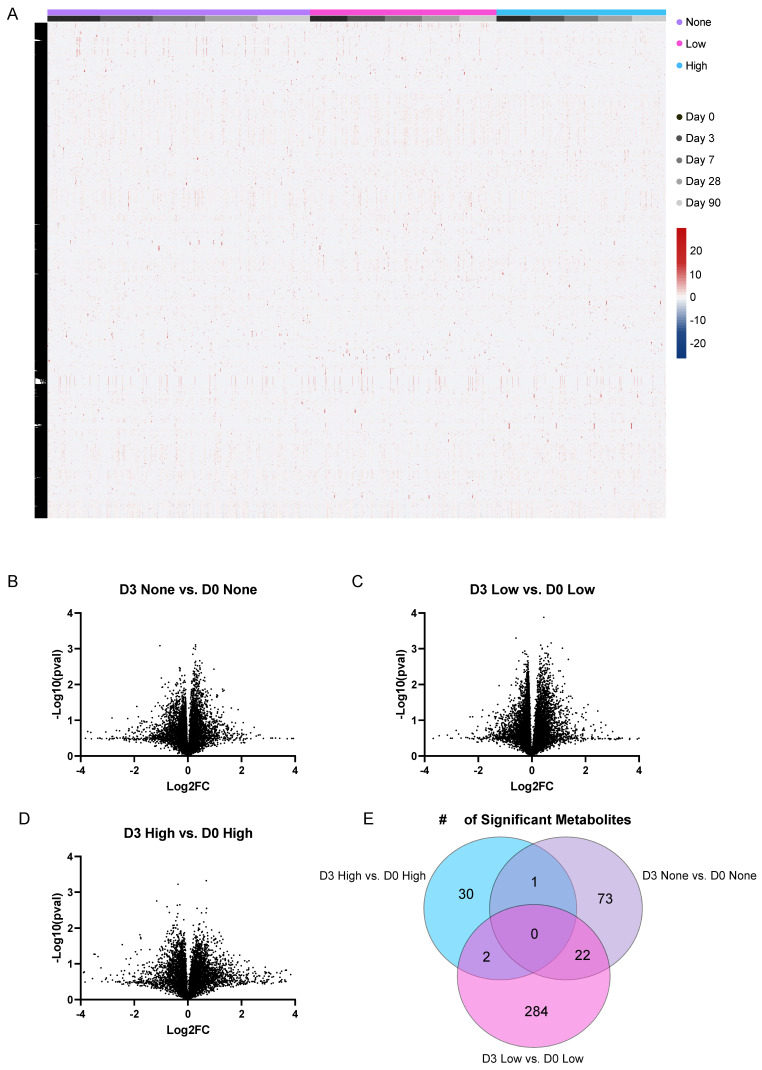
Summary of metabolomics data and subject response by seroconversion. (**A**) Overall semi-supervised hierarchical clustering for the metabolomics analysis; *n* = 895 samples analyzed with the relative quantification of 15,903 putatively identified metabolites. (**B**–**D**) Volcano plot representation of subject-specific metabolite fold change between Day 3 post-vaccination vs. Day 0 pre-vaccination. The x-axis represents the fold change and Log2, while the Y-axis represents statistical significance (t-test, two-tailed, equal-variance, uncorrected). (**E**) Venn diagram of overlapping significant metabolites between the three D3 vs. D0 comparisons of the High-, Low-, and None-responders, respectively.

**Figure 3 viruses-15-00242-f003:**
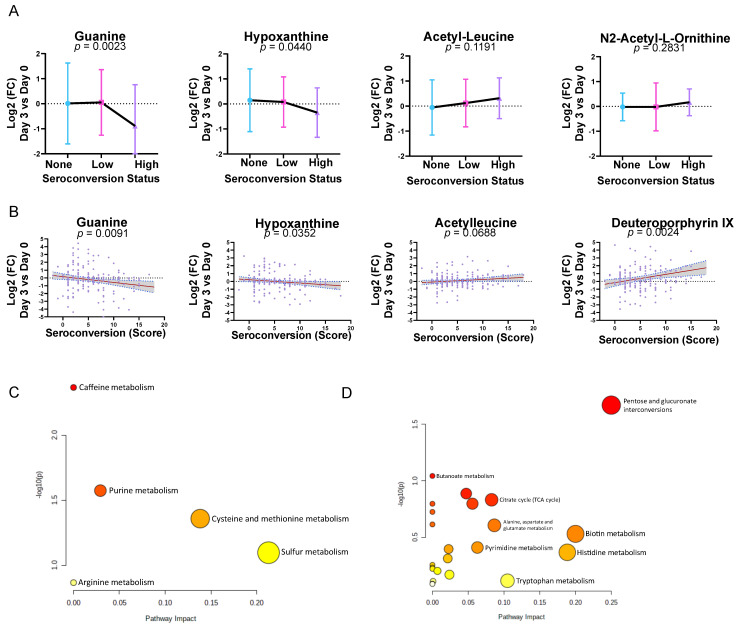
Metabolites associated with the response of the High-responders to vaccination. (**A**) Subject-specific changes in metabolites with respect to the seroconversion status on Day 3 vs. Day 0. Error bars represent the standard deviation. (**B**) Overall correlations of metabolite markers with the subject seroconversion score, *n* = 179, where the *p*-value represents the significance of the non-zero slope for linear regression. The red line represents the best-fit regression, and the grey area represents the 95% confidence interval for the line of regression. (**C**,**D**) Metabolic pathway analysis of the High-responders and None-responders, respectively. The x-axis represents the pathway impact (% of pathway covered), and the y-axis represents the statistical significance.

**Figure 4 viruses-15-00242-f004:**
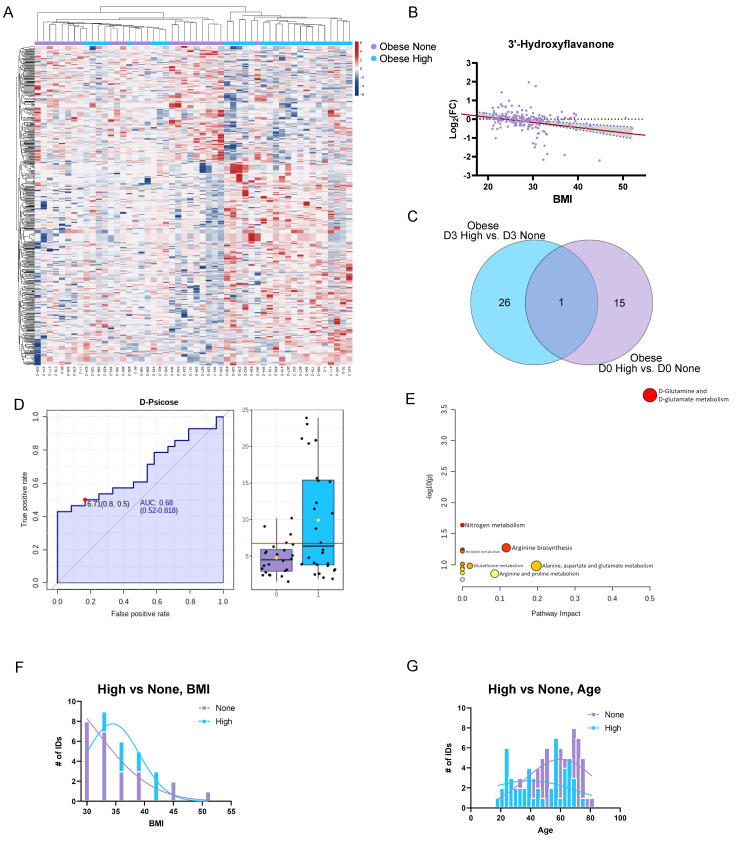
Differential metabolic profiles of the obese High- and None-responders. (**A**) Unsupervised hierarchical clustering of the metabolite profiles of the obese High-responders and obese None-responders, with *n* = 523 putatively identified metabolites with a raw *p*-value of <0.01 (t-test). (**B**) Scatter plot correlation of subject BMI with the D3/D0 change in the relative metabolite levels. The red line represents the best-fit regression, and the grey area represents the 95% confidence interval for the line of regression. (**C**) Venn diagram of overlapping significant metabolites between the obese High-responders and obese None-responders on Day 3 (D3) and also on Day 0 (D0). (**D**) Receiver operator characteristic curve (ROC curve) for the putatively identified D-Psicose metabolite discriminating the obese High-responders (blue) from the obese None-responders (purple). AUC = area under the curve. The red line indicates the test threshold for group discrimination. (**E**) Metabolic pathway analysis of the predictive (baseline) metabolites between the obese High-responders and obese None-responders. (**F**) BMI histogram for the High-responders and None-responders. (**G**) Age histogram for the High-responders and None-responders.

**Figure 5 viruses-15-00242-f005:**
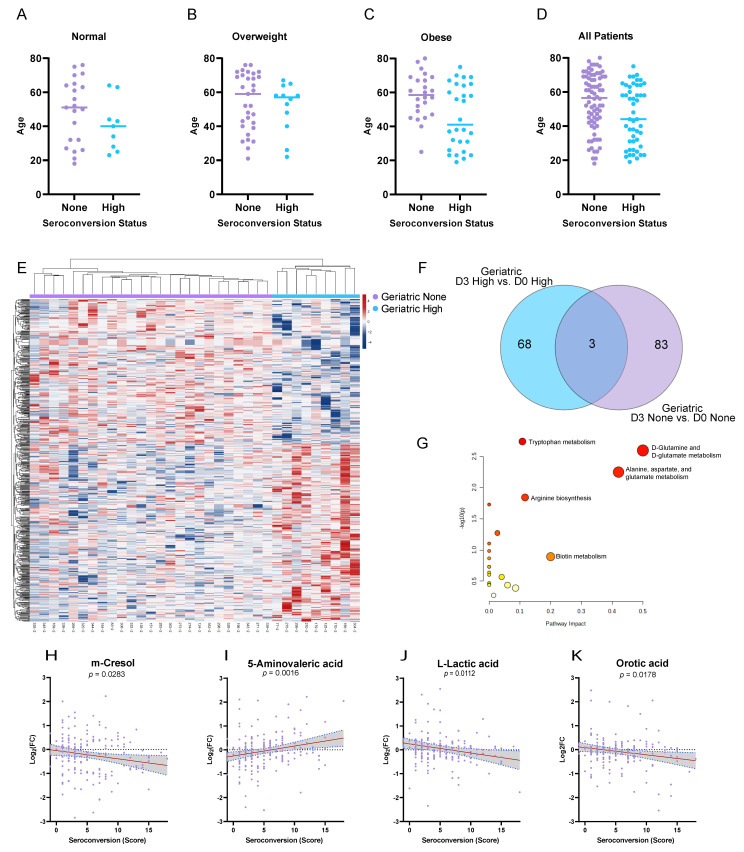
Differential metabolic profiles of geriatric High- and None-responders. (**A**–**D**) Scatter distribution of the ages of all the cohort subjects with respect to the BMI category. (**E**) Unsupervised hierarchical clustering of the metabolite profiles of geriatric High-responders and geriatric None-responders, with *n* = 624 putatively identified metabolites with a raw *p*-value of < 0.05 (t-test). (**F**) Venn diagram of overlapping significant metabolites between geriatric High-responders and geriatric None-responders on Day 3 (D3) and also on Day 0 (D0). (**G**) Metabolic pathway analysis of the uniquely changed metabolites among the geriatric None-responders. (**H**–**K**) Scatter plot correlations of the subject seroconversion score with the D3/D0 change in the relative metabolite levels. The red line represents the best-fit regression, and the grey area represents the 95% confidence interval for the line of regression, *n* = 179 for all the correlations.

**Figure 6 viruses-15-00242-f006:**
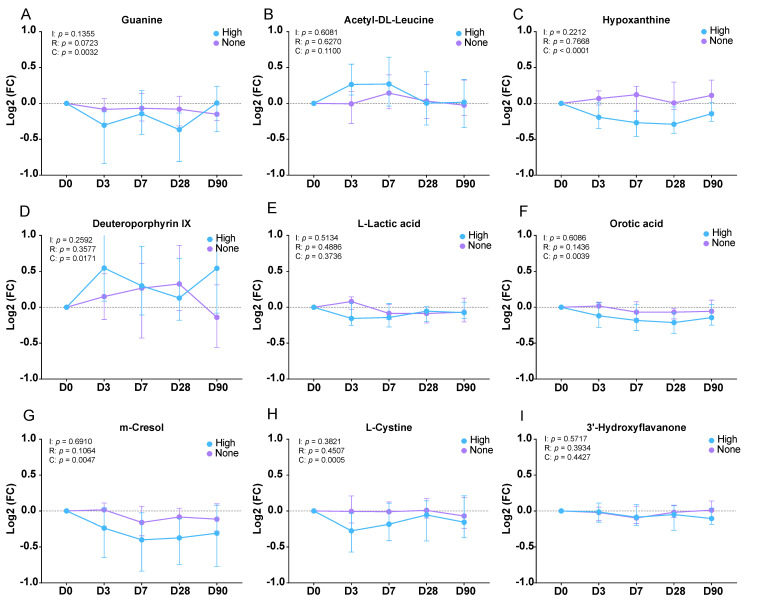
Metabolite time-course dynamics of the IAV vaccine response. (**A**–**I**) Line plot of the relative change in the subject urine metabolite levels with respect to the baseline over the 90-day time-course. For each metabolite, error bars represent the standard deviation of the fold change compared to the baseline, and the Y-axis represents the Log2 values. Each metabolite was analyzed in a two-way ANOVA. I = interaction of the two factors, R = row factor, days post-vaccination, C = column factor, seroconversion status (High-responder or None-responder).

## Data Availability

Not applicable.

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
