# Peer review of "Urine Metabolome Dynamics Discriminate Influenza Vaccination Response"

_viruses, 2023, doi:10.3390/v15010242_

Round 1

Reviewer 1 Report

This paper studies the metabolic profiles in the urine of individuals that have been vaccinated against influenza A viruses (IAV) and correlates these profiles to seroconversion and protection against IAV. The authors focus on age and BMI status as possible factors influencing the post-vaccination seroconversion.

A lot of the results seem to be statistically insignificant, which is not clearly shown in the figures, but mentioned in the text. The pathway analyses are the focus of the paper based on its title, but it does not seem that some of these results are supported by statistics. I understand that it might be difficult to see clear statistical differences given how many uncontrollable variables there are in an observational study like this, and that there is a noticeable trend in the metabolites analyzed. However, these results could also be seeing by chance in a specific geographical location or population habits, culture, etc. In summary, the experimental design and the volunteer selection could have been better explained. If the selection of participants is somehow biased, then the conclusions from this paper could be misleading. 

The manuscript is a heavy read and has several figures that need to be better explained so that the reader can keep up with the authors’ findings. In addition, most of the figures are very low quality, which makes their evaluation difficult. 

More detailed comments are listed below.

Line 97: How were the volunteers selected for the study? I wonder if there is any sort of sampling bias regarding economic status, for example. Poor participants would have a different diet, access to public health, exposure to neighborhoods with greater environmental hazards, they may have jobs that imply a greater risk, and so on. It would be relevant to inform the reader about how the experiment was advertised and how the participants enrolled in the experiment.

Lines 106-108: What was the geographic location of the participants? Is the ethnicity of the participants representative of the city or state where they belong to? If not, could this represent a sampling bias as mentioned above?

Line 135: Using a less stringent cutoff of P<0.05, what was the overlap between none, low, and high responders? On another note, I appreciate Figure 2E. It is interesting to see that none and low responders have more metabolites in common (n=22) compared to high responders (n=1 and n=2, respectively).

Lines 136-138: I am not sure what the authors mean in the sentence starting in “…, indicating”. What is the relationship between the lack of overlap in metabolites between the three groups and participation in the study?

Lines 315-316: It seems logical and almost common knowledge that obesity and ageing are correlated with poor immune responses. If I understood correctly, the authors arbitrarily chose age and BMI as risk factors for poor seroconversion and decided to focus on these in their analyses. This question (whether age and BMI are determinants of poor seroconversion) was already answered when the experimental design was made.

Lines 316-319: I thought the tryptophan metabolism pathway turned out not significant after Holm’s correction (line 256).

Lines 333-335: One could hypothesize that elderly people might consume more low-calorie sweeteners than younger people who look for healthier sugar alternatives. Likewise, obese people will likely consume more low-calorie sweeteners than people with their BMI within normal range. I do not see these metabolites as having a potential role in vaccine response prediction.

Lines 353-363: What do the p values on Figure 3A (p=0.12 for acetyl-leucine and p=0.28 for deuteroporphyrin) mean? It does not seem that none, low and high seroconversion groups are different by looking at the standard deviations of these two metabolites. Having that said, is it fair to say these metabolites can be used as seroconversion markers? You are using the correlation statistical significance to assess this relationship, but there is no difference between the tested groups as per the ANOVA p values shown in Figure 3A.

Lines 391-394: I appreciate the author’s comment. I agree that HI is not a perfect measure of seroconversion, but it is an excellent tool for detecting neutralizing antibodies. I also agree that previous immunization and cross-reactivity could interfere in your analysis, and perhaps controlling sampling for previous vaccination (like for other factors shown in Figures 1B-E) would have helped. However, I think this brings a real-life background to your analysis, especially because many people might have had undiagnosed flu in their lives and that would not be accounted for. Another thing to consider is the original antigenic sin, in which people might be poor responders depending on the epitopes of flu they were previously exposed to.

Lines 387-411: I really appreciate this part of the discussion where the authors acknowledge many of the issues I had with the paper.

Figure 1A: Was seroconversion only measured on day 28? It would have been interesting to see a paired HI test at 90 days post-vaccination.

Figures 1B-E: It would be easier if the authors added percentages to the pie graphs.

Figure 1H: What does “others” include in terms of ethnicity?

Figure 2A: The image quality is too low; hard to see any clustering.

Figure 4: The legend of this figure is very poorly elaborated. For example, is 4G an age histogram for obese high and none responders like mentioned in lines 221-224? Please elaborate on the entire figure explanation, all panels should be self-explanatory.

Figure 5E: The image quality is too low; hard to see the legend.

This reviewer did not have access to any supplementary data. The externally provided link to access the files was not publicly available.

Reviewer 2 Report

The efficacy of the current influenza virus vaccines varies due to molecular differences between the vaccine strain and the circulating strain, but also person to person variation in seroconversion. Rodrick et al analyzed urine samples from 179 volunteers to identify small molecules levels following vaccination and explore if they play a role in vaccine efficacy and/or could be predictive markers for vaccine efficacy. The authors accounted for differences in sex, BMI and age, but the majority of volunteers was Caucasian. The authors find that a robust seroconversion is associated with changes in the metabolite profile. In particular, the authors find that purine, Glutamic acid and Pyroglutamic acid, D-Psicose, Acetyl-Leucine and Deuteroporphyrin IX levels are interesting markers for poor or robust seroconversion in the groups that they studied, or in the case of the latter two, potentially even across the different groups. In addition, the authors find that age is the best predictor of poor seroconversion, which is in line with previous studies. Future studies to confirm the role of the metabolites are proposed by the authors in the discussion. Overall, the manuscript is well-written, the data well-presented and the study robustly analyzed. I also appreciate that the authors are cautious with their data interpretation, in line with the design of the study. There are a few minor typos in the manuscript, but those should be easy to filter out in the editing stage.

Round 2

Reviewer 1 Report

Even though the authors clarified some of my questions and comments from the first review, the methods (i.e., selection of participants) and results are still not able to support the conclusions.

I am not sure how the authors can confidently say “the ethnicity of participants is in a similar proportion as the background” if they had 86% of Caucasian individuals in their study while the Caucasian population in Athens is 58%.

Regarding Figure 2E, I am not sure why the authors are focusing on pathways that none, low, and high responders have in common. Shouldn’t you be focusing on the metabolites that do not overlap between groups to be able to use them as markers of better or worse response? Likewise, the authors are trying to find metabolites that correlate with high BMI and age, but they focus on metabolites that high and none responders have in common (e.g., D-Psicose, lines 211-214).

Most if not all of their conclusions are based on cellular pathways where we cannot see statistical differences. The authors justify their results saying the trends are relevant and the lack of statistical differences are due to a small sample size (n=179 individuals). To me, these results are not robust enough to support their conclusions, and the metabolites could have been found by chance.
